# Cryo-EM structures reveal how phosphate release from Arp3 weakens actin filament branches formed by Arp2/3 complex

Sai Shashank Chavali[1], Steven Z. Chou [2,5], Wenxiang Cao [1], Thomas D. Pollard [1,2,3,4] ✉, Enrique M. De La Cruz [1] ✉ & Charles V. Sindelar [1] ✉

Arp2/3 complex nucleates branched actin filaments for cell and organelle movements. Here we report a 2.7 Å resolution cryo-EM structure of the mature branch junction formed by *S. pombe* Arp2/3 complex that provides details about interactions with both mother and daughter filaments. We determine a second structure at 3.2 Å resolution with the phosphate analog $BeF_x$ bound with ADP to Arp3 and ATP bound to Arp2. In this ADP-$BeF_x$ transition state the outer domain of Arp3 is rotated 2° toward the mother filament compared with the ADP state and makes slightly broader contacts with actin in both the mother and daughter filaments. Thus, dissociation of $P_i$ from the ADP-$P_i$ transition state reduces the interactions of Arp2/3 complex with the actin filaments and may contribute to the lower mechanical stability of mature branch junctions with ADP bound to the Arps. Our structures also reveal that the mother filament in contact with Arp2/3 complex is slightly bent and twisted, consistent with the preference of Arp2/3 complex binding curved actin filaments. The small degree of twisting constrains models of actin filament mechanics.

Arp2/3 complex is composed of seven protein subunits and nucleates the formation of actin filament branches on the sides of preexisting "mother" actin filaments[1]. Elongation of these branched actin filament networks powers eukaryotic cell and organelle movements[2]. Like actin, Arp2/3 complex is an ATPase. ATP hydrolysis and release of the γ-phosphate from Arp2 and/or Arp3 regulate the turnover and mechanical stability of branched filament networks[3–7].

High resolution crystal structures of Arp2/3 complex have been available since 2001[8–10]. Advances in electron microscopy have improved the resolution of structures of branch junctions from 25 Å[11] to 9 Å[12], 3.9 Å[9,13] and 3.5 Å[8]. In all these studies the branches were aged, so the ATP bound to the actin and Arps was hydrolyzed and the γ-

phosphate dissociated from the active sites (except for ATP bound to Arp2 in the *S. pombe* Arp2/3 complex[8]). These structures established the overall architecture of the branch junction and the conformational changes in Arp2/3 complex during branch formation.

Pico-newton forces applied with hydrodynamic flow in micro-fluidic chambers dramatically increased the rate that branches dissociate from mother filaments[7]. Newly formed branches are 20 times less sensitive to force than mature branches that have time to dissociate phosphate from the ADP-$P_i$ intermediate in the active sites of the Arp2/3 complex. Similarly, the high affinity phosphate analog, $BeF_x$, also stabilizes branch junctions. These observations raised the question of how the presence of the γ-phosphate in the active site

[1]Department of Molecular Biophysics and Biochemistry, Yale University, PO Box 208103 New Haven, CT 06520-8103, USA. [2]Department of Molecular Cellular and Developmental Biology, Yale University, PO Box 208103 New Haven, CT 06520-8103, USA. [3]Department of Cell Biology, Yale University, PO Box 208103 New Haven, CT 06520-8103, USA. [4]Department of Molecular and Cell Biology, University of California, 638 Barker Hall, Berkeley, CA 94720-3200, USA. [5]Present address: Department of Molecular Biology and Biophysics, University of Connecticut Health Center, Farmington, CT 06030, USA. ✉e-mail: thomas.pollard@yale.edu; enrique.delacruz@yale.edu; charles.sindelar@yale.edu

strengthens the branch junction, but the available structures do not provide the answer.

We report two high resolution cryo-EM structures of actin filament branch junctions formed by *S. pombe* Arp2/3 complex. A 2.7 Å resolution structure with ADP bound to the actin subunits in the mother and daughter filaments, ADP bound to Arp3 and ATP bound to Arp2 reveals several functionally important parts of Arp2/3 complex absent from lower resolution structures. This structure serves as a reference standard for the mature branch junction. A second structure at 3.2 Å resolution has the phosphate analog BeF$_x$ bound with ADP in the active sites of Arp3 and ATP bound to Arp2. Compared with the ADP reference structure, the branch junction with ADP-BeF$_x$ has a 2° rotation of the outer domain of Arp3 places subdomain 2 (SD2) 0.8 Å closer to the mother filament and buries more surface area between Arp3 and actin subunits in both the mother and daughter filaments. These differences between the ADP-BeF$_x$ and the ADP-structures are expected to be associated with phosphate release and may contribute to reducing the stability of aged branch junctions[7].

## Results

### Cryo-EM structure of a mature Arp2/3 complex branch junction at 2.7 Å resolution

We used cryo-electron micrographs of ~400,000 branch junctions in 16,000 images to reconstruct Arp2/3 complex associated with mother and daughter actin filaments at 2.7 Å resolution (Fig. 1; Table 1). This structure confirms the overall architecture of the branch junction and also reveals interactions that stabilize the branch junction not seen at lower resolution[8–10].

The higher resolution allows unambiguous modeling of the sidechains of most residues in 8 actin subunits of the mother filament, 2 actin subunits of the daughter filament, and all 7 subunits of Arp2/3 complex. All 8 actin subunits and Arp3 have well defined densities in their active sites for ADP and a coordinated Mg$^{2+}$ ion. The "backdoor" for phosphate release is closed by a hydrogen bond between R177 and N111 in all mother filament actin subunits. The map shows clearly that Arp2 has bound nucleotide triphosphate that is consistent with Mg$^{2+}$-ATP, although incomplete occupancy of the γ-phosphate suggests that partial hydrolysis and phosphate dissociation may have occurred[8].

### Features of the interactions between Arp2/3 complex and the mother filament

All subunits in the Arp2/3 complex, except for Arp2 and ARPC5, contact the mother filament directly. The following paragraphs focus on herein identified features of these interactions that were not resolved in previous, lower resolution structures.

At 2.7 Å resolution, Arp3 D-loop residues 39-57 are disordered as in previous structures, but we also visualize weak density for the D-loop C-terminal region (residues 58-67) in the branch junction (Fig. 1b). The Arp3 D-loop density indicates that residues 58-67 interact with the C-terminal α-helices of ARPC2 and ARPC4 and subdomain-3 (SD3) of actin subunit M4 of the mother filament. Arp3 subdomain-2 also makes multiple interactions with subdomain 1 (SD1) of actin subunit M3, including salt bridges between Arp3 R95 and actin E364 and Arp3 R225 and actin D363 (Supplementary Fig. 1A). Additionally, Arp3 Y218 is within hydrogen bonding distance of actin Q360 (Supplementary Fig. 1A).

At higher resolution the sidechain densities for ARPC1 residues 318-330 of the insert helix are sharper, enabling accurate placement of rotamers (Fig. 1c). The ARPC1 insert helix nestles between the C-terminal tail of ARPC2 and SD3 at the barbed end of mother filament actin subunit M3. The interactions are largely hydrophobic, including contacts of ARPC1 F317 with actin I345, ARPC1 F324 with actin L349 and other interactions of ARPC1 residues M327 and L320 and M3 residue I341 (Supplementary Fig. 1B). ARPC1 R321 forms an electrostatic

interaction with ARPC2 E303 and a hydrogen bond with M3 S348 (Supplementary Fig. 1C). The high-resolution map lacks density for the linker between the ß-propeller and N-terminus of the insert helix, further evidence that it is disordered.

The higher resolution ARPC2 structure clarifies the sidechain conformations at the C-terminus of ARPC2 and their interactions with ARPC1 (Fig. 1d). ARPC2 also interacts with two mother filament actin subunits (M3 and M5) as well as Arp2 and Arp3. Salt bridges form between ARPC2 K194 and actin M5 E125, ARPC2 R205 and M5 E117, and ARPC2 D202 and M5 K118 (Supplementary Fig. 1D). Residues R306 and K307 in the C-terminal tail of ARPC2, interact with SD1 of actin M3 and also bridge between the N-terminus of actin M3 and Arp2 subdomain 4 (residues 241-247) (Supplementary Fig. 1C).

ARPC3 makes major contacts with the inner domain of Arp3 (buried surface 898 Å$^2$) and the D-loop of Arp2 (buried surface 639 Å$^2$) and buries a modest surface area of 176 Å$^2$ with mother filament subunit M1. The backbone oxygen of ARPC3 residue R94 forms a hydrogen bond with actin M1 residue Q354.

ARPC4 makes numerous contacts with the mother filament (Supplementary Fig. 1E). The C-terminal α-helix (residues 128-165) of ARPC4 interacts with SD1 of mother filament subunit M5 (buried surface 764 Å$^2$) through salt bridge and hydrogen bond interactions. ARPC4 R158 forms salt bridges with residues E83 and D80 in SD1 of M5, which are reinforced by a hydrogen bond network among ARPC4 N154 and actin M5 residues E83 and T126. Furthermore, ARPC4 forms two additional salt bridges with SD1 of actin subunit M5: between ARPC4 R55 and M5 E99; and between ARPC4 residue M5 E100.

The extended N-terminus of ARPC5 (residues 2-27) is better resolved than in previous structures, revealing sidechain interactions with subdomains 3 and 4 of Arp2 (Supplementary Fig. 1F). Salt bridges form between ARPC5 R4 and Arp2 E316 and ARPC5 E13 and Arp2 K318. ARPC5 residue V8 makes a hydrophobic contact with Arp2 F322.

The mother filament is bent where it associates with Arp2/3 complex in the ADP branch junction (Fig. 2). Arp2/3 complex binds on the convex side of the bent mother filament. To measure bending, we generalized methods to quantify bending and twisting of DNA in atomic structures[14] for application to other molecular assemblies (Fig. 2). This analysis revealed a net end-to-end bending of ~2.1° over 4 subunits (M2-M5; Fig. 2c, h).

The geometry of this bend indicates a significantly strained, but still thermally accessible, conformation of the mother filament. The majority of filament bending (~1°) occurs between subunits M2 and M3, while the remainder is distributed between subunits M3–M6. Corresponding radius of curvature values between adjacent, bent subunit pairs are in the range of ~0.15–0.3 μm ($R = s/\Delta\theta_i$, where $s = 2.75$ nm is the segment length and $\Delta\theta_i = $ ~0.005–0.015 in radians is the end-to-end angular deflection between two subunits). This is associated with a total of ~0.9 $k_B T$ in elastic strain energy ($E_s$) within the bent subunits M1 to M4. We estimated the energy per bent subunit pair as $0.5 L_B s/R_i^2\, k_B T = 0.5 L_B \Delta\theta_i^2/s\, k_B T$, where $L_B$ is the bending persistence length ≈10 μm[15–17], and estimated bending values of 1.0° for M2-M3, 0.7° for M3-M4, and 0.4° for M4-M5.

Our analysis of the filament geometry also reveals minor twisting of the mother filament compared with a helically symmetric actin filament (Supplementary Fig. 2A). The twisting varies significantly from one subunit to the next, deviating in a range of +/– ~0.5° from the symmetric value (166.7°).

To further assess filament twisting, we estimated the excess end-to-end twist, or 'cumulative net twist difference'[18] from the average twist of the straight filaments, over the length of the filament (Fig. 2j). Excess end-to-end twist values are calculated from the twist values in Supplementary Fig. 2A according to the description given in the Supplementary Notes and are plotted using the same line styles. The cumulative net twist difference is highly sensitive to small changes

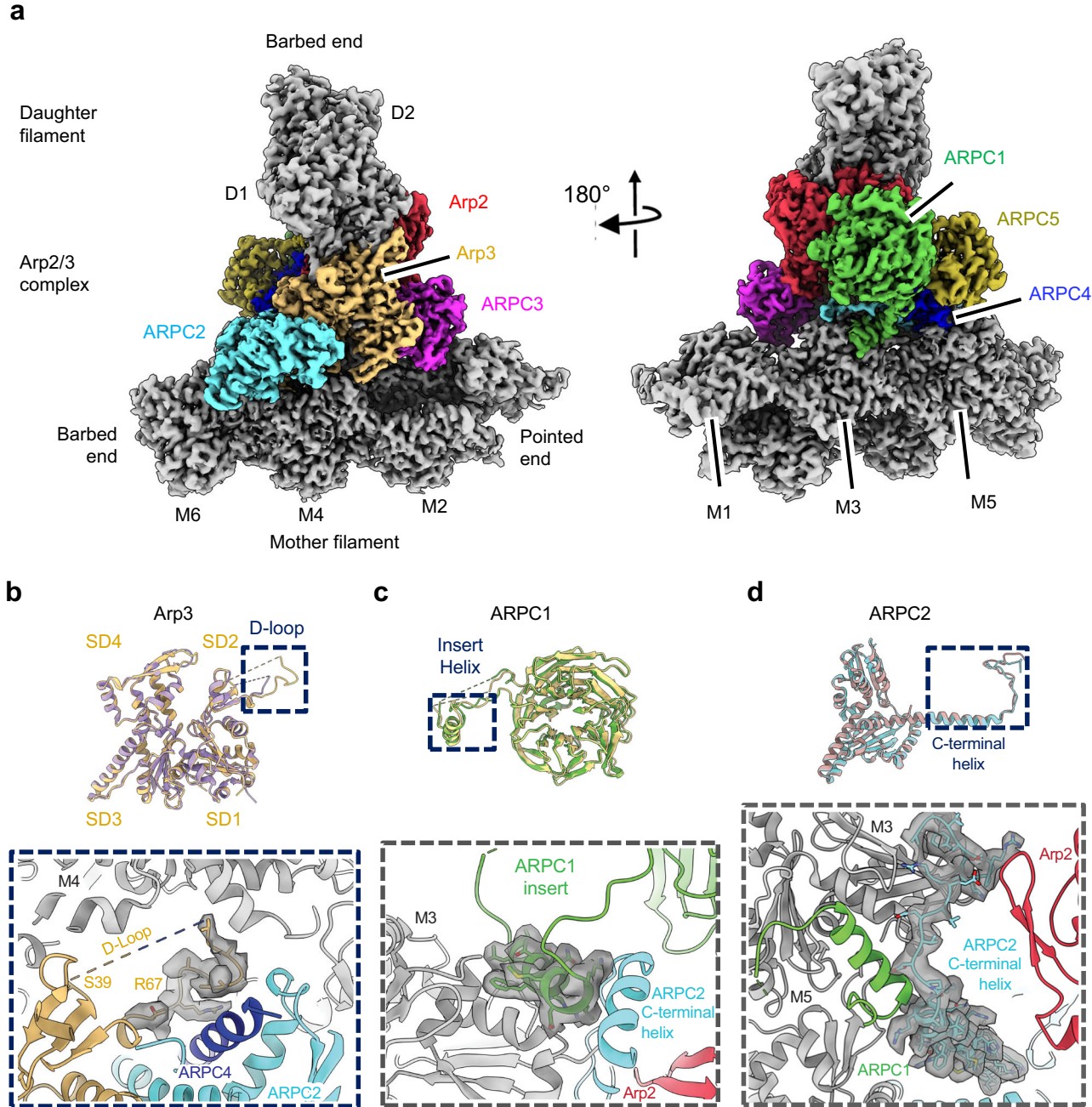

**Fig. 1 | Features in the 2.7 Å resolution reconstruction of the ADP-bound branch junction. a** Electron potential map of the 2.7 Å reconstruction of a branch junction with Arp2/3 complex and the actin subunits in the ADP-state. **b**–**d**, Overlayed models of the 2.7 Å and 3.5 Å (Chou et al. PDB 8E9B)[8] junction reconstructions. Upper part of each panel has overlays of the two models. The lower part shows details of herein resolved structures. **b** The 2.7 Å map of Arp3 allows placement of backbone atoms of the D-loop (residues 58 – 67), which were missing at lower resolution. D-loop residues 39-57 are disordered in both structures as indicated by

the breaks in the backbone. The D-loop of Arp3 interacts with ARPC2, the ARPC4 C-terminal helix and mother actin filament subunit M5. **c** The conformations of the insert helix of ARPC1 (residues 318-330) differ slightly in the two structures. The insert helix bridges the mother actin filament M3 and the C-terminal tail of ARPC1. **d** The map of ARPC2 has unambiguous densities for the side chains densities of the C-terminal helix and extended tail (residues 270-320). The tail residues (300-317) bridge mother actin filament (subdomain 1 and N-terminus of M3) and the subdomain 4 of Arp2.

over a span of many subunits, because the measurement error stays constant with increasing filament length[18]. Most of the twisting is localized near subunits M2-M4 where bending is observed, and the cumulative twist difference is ~ +0.9° over the full range of the bent filament segment (subunits M1-M6; Fig. 2j). The positive sign of the cumulative twist difference indicates that the filament is slightly overtwisted compared to the straight, symmetric case. The small torsional elastic strain energy ($E_s \ll k_B T$, calculated using a filament torsional persistence length $L_T$ value of 10 μm[15]) associated with these bending

deformations is consistent with model predictions of twist-bend coupling in actin filaments[19].

For comparison, we also computed the twist (Fig. 2k, Supplementary Fig. 2B) associated with bending of two bare actin structures in ADP-$P_i$ and ADP nucleotide states[20]. As with the mother filament structures analyzed here, there is a small but significant net overtwist associated with filament bending (Fig. 2k). This value is more than an order of magnitude smaller than that reported from the analysis of these same bare filament structures[20].

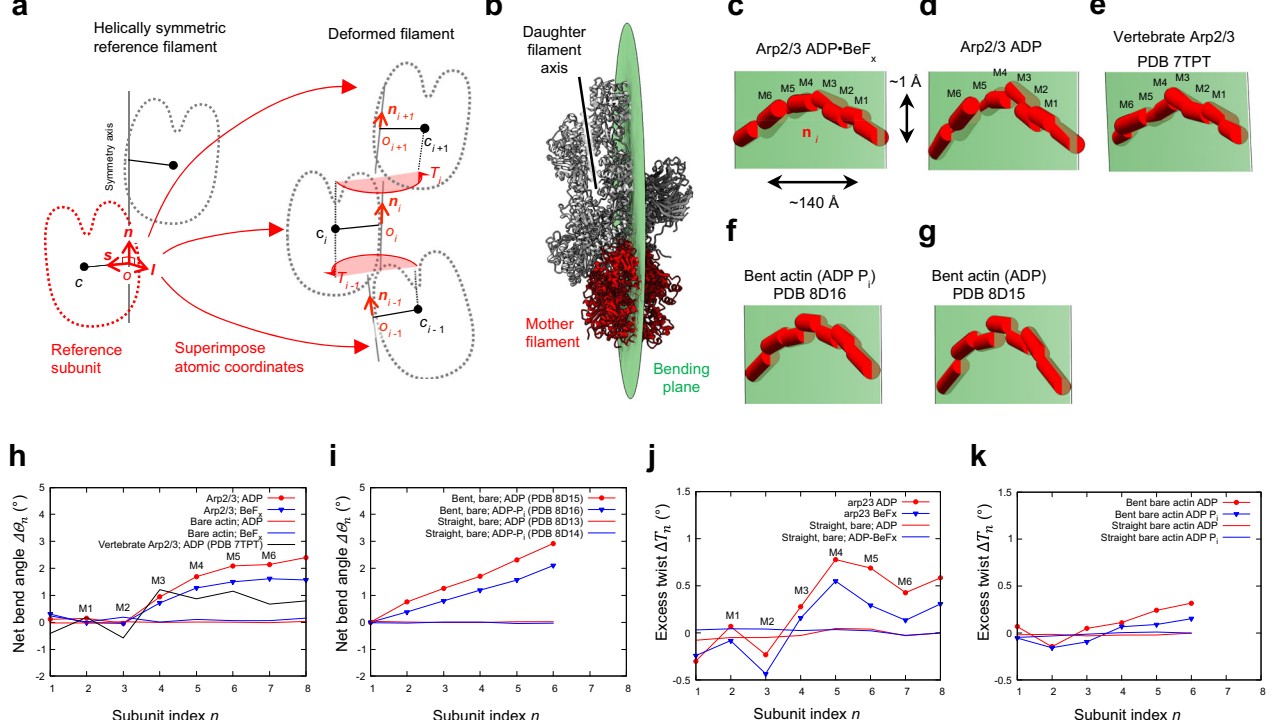

**Fig. 2 | Bending and twisting of mother filaments in branch junctions.**
**a** Definition of the subunit coordinate frame and use of this frame to follow the path of a distorted filament[14] (see Methods). **b** Ribbon diagram of the ADP branch junction viewed from the pointed end with a 10° offset along the axis of the mother filament (red) with Arp2/3 complex and daughter filament in gray. The green disk is the plane fit by least-squares superposition of the subunit origin coordinates $o_i$ along the bent mother filament through Arp2/3 complex, roughly parallel with the daughter filament axis (solid black line). Arp2/3 complex is on the convex side of the mother filament (see panels **c**–**d**). **c**–**g** Paths of subunits in bent actin filaments, viewed from the pointed end. Note that the vertical deflection of the paths (vertical scale bar in **c**) is highly exaggerated compared with the end-to-end distance (horizontal scale bar in **c**). Subunits are represented by coordinate frame vectors $\mathbf{n}_i$, depicted as red three-dimensional rods. Semitransparent green surfaces are bending planes through the path centers $o_i$ (fit by least squares). The staggered frame vectors in bent filaments (**c**–**g**) but not straight actin filaments, indicate shearing between neighboring subunits compared with the straight reference structure. **c, d** Mother filament subunit paths in our branch junctions with ADP BeF$_x$- and ADP-Arp2/3 complex. **e** Path of mother filament subunits in branch junctions with bovine Arp2/3 complex (PDB 7TPT) is within 5° of those in **c**, **d**. **f, g** Paths of actin filament subunits in bent actin filaments with ADP-P$_i$ (PDB 8d16) and ADP (PDB 8d15) subunits[20]. **h–i** In-plane bending angles between subunits $n$ and $n+1$ in actin filaments. **h** Mother filaments in branch junctions (**c**–**e**) have a one-degree bend between subunits M3 and M5. **i** Bent ADP-P$_i$- and ADP-actin filaments[20] have approximately constant curvatures. **j, k** Excess end-to-end twist, or 'net twist difference'[18] between the first and $n$th subunits (inclusive) as a function of $n$. (See Supplemental Fig. 2 and supplemental notes) **j** ADP BeF$_x$- and ADP-branch junctions. **k** Bent ADP-P$_i$- and ADP-actin filaments[20]. Source data for panels **h**–**k** are provided as a Source Data file.

## Interactions between Arp2/3 complex and daughter filament

The high-resolution map reveals features of the interaction of the pointed end of daughter filament subunit (D1) with the barbed end of Arp3. At lower resolution[8] the D-loop of actin subunit D1 encircles Y200 of Arp3 (corresponding to actin Y169), but the density of the sidechain of D1 M44 was weak. Our map has a well-resolved densities for the sidechains of D1 M44 and M47, which make hydrophobic interactions with Arp3 (Fig. 3c). The sidechain of M44 fits in a hydrophobic pocket above the Y200-loop of Arp3. Additional interactions between the barbed end of Arp3 and pointed end of D1 are a hydrogen bond between the backbones of D1 M44 and Arp3 G199, and a salt bridge between R62 in SD2 of D1 and D321 in SD3 of Arp3.

The D-loop of daughter filament subunit (D2) wraps around V169 (corresponding to actin Y169) in the barbed end groove of Arp2. The side chain of actin subunit D2 M44 occupies a hydrophobic pocket above the V169-loop and the hinge-helix of Arp3. Further interactions between actin D2 and Arp2 include a backbone hydrogen bond interaction between D2 residue with Arp2 S168, two hydrogen-bonds between D2 D-loop residue H40 with H173 in SD1 of Arp2 and the Nε atom of D2 Q49 with the oxygen of S168 in SD1 of Arp2 and two salt bridges between R62 in SD2 of D2 with Arp2 D288 (corresponding Arp3 D321), and between E241 in the SD4 of D2 with R332 in the SD3 of Arp2.

## Cryo-EM structure of Arp2/3 complex with ADP-BeF$_x$ in the actin filament branch at 3.2 Å resolution

We stabilized the structure of the branch junction, corresponding to an ADP-P$_i$ intermediate state of the Arp2/3 complex, by including 2 mM BeSO$_4$ and 10 mM NaF in the buffer during daughter filament formation[7].

We reconstructed the branch junction with ADP-BeF$_x$ from ~200,000 particles in 8500 cryo-electron micrographs. The quality of the 3.2 Å resolution map allowed for the unambiguous placement of most residues in 8 actins and Arp2/3 complex in an atomic model with the same number of residues as the ADP-structure, except for residues 40–64 in the D-loop of Arp3, which were poorly resolved. The branch structure with ADP-BeF$_x$ includes all high-resolution features described for the branch structure with ADP, including specific interactions between Arp2/3 complex with the mother and daughter filaments that stabilize the branch junction.

## Nucleotides bound to actin and Arp2/3 complex

Density for BeF$_x$ at the γ-phosphate position next to Mg$^{2+}$-ADP is seen in 5 mother filament actin subunits and 1 daughter filament actin subunit (Supplementary Fig. 3). The density for BeF$_x$ is weak in the remaining 2 actin subunits (D2 and M6), likely due to their location at the volume periphery where the BeF$_x$ would be less

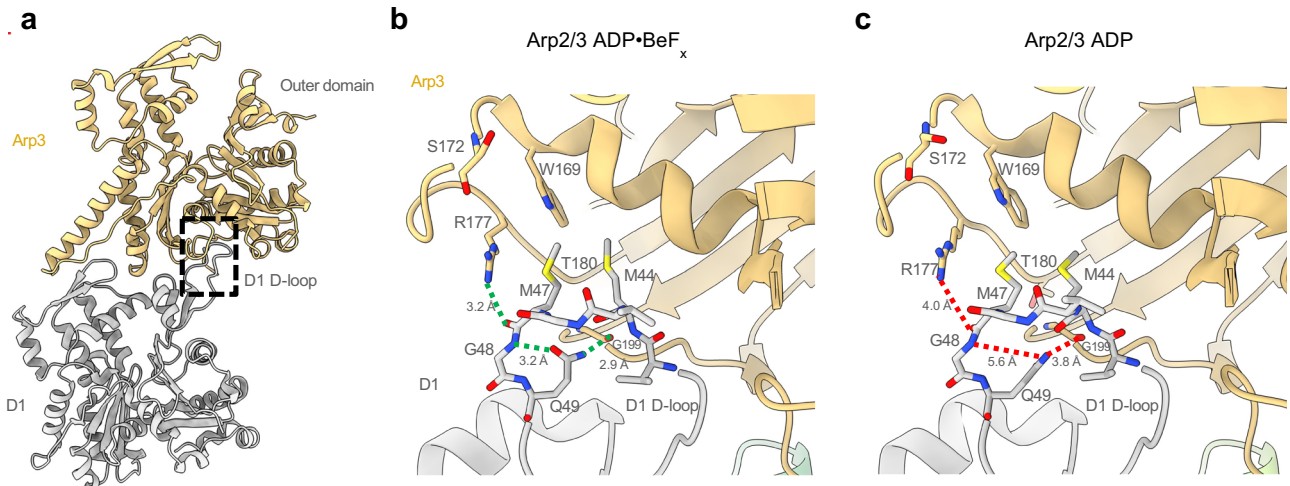

**Fig. 3 | Models comparing interactions of Arp3 with daughter filament subunit D1 in branch junctions with ADP-BeF$_x$ or ADP. a** Ribbon diagrams of Arp3 and D1 from the branch junction with BeF$_x$ nucleotide. A box highlights part of the interface between daughter subunit D1 and Arp3 shown in more detail in (**b**) and (**c**). **b** Ribbon diagrams with stick figures of residues 44–49 of the D-loop of actin subunit D1 in the ADP-BeF$_x$ state. The interaction of the D-loop with Arp3 is stabilized by hydrophobic interactions including actin D1 residues M44 and M47 with Arp3 and multiple hydrogen bonds: Arp3 R177 with the backbone oxygen of actin M47; and the side chain of actin Q49 with the backbones of both its own G48 and Arp3 G199. **c** In ADP state the outer domain of Arp3 twists by 2°, so the hydrogen bonding interactions are further apart, thus weakening the Arp3-D1 interface.

well resolved. The map of Arp3 has clear density corresponding to BeF$_x$ in the ATP γ-phosphate position (Fig. 4a). This density is absent in the ADP-Arp3 map (Fig. 4b). Similar to the ADP branch junction structure, the reconstruction of the branch junction prepared with BeF$_x$ has strong density for γ-phosphate in the active site of Arp2 (Fig. 4c), confirming that Arp2 of *S. pombe* Arp2/3 complex does not hydrolyze its bound ATP rapidly during branch formation[8].

### Arp3 phosphate release modulates interactions with the mother and daughter filaments

In the branch junction with ADP-BeF$_x$ bound to Arp3, Arp3 is slightly more flattened than Arp3 in branches with bound ADP owing to a difference of ~2° in the angle between outer and inner domains (Fig. 4d–f). This outer domain rotation (Fig. 4d–f), which would be associated with transition from the ADP-BeF$_x$ state to the ADP state, moves the D-loop of Arp3 0.8 Å away from the mother filament (Fig. 5).

**Mother filament bending.** Mother filaments in branches with ADP-BeF$_x$ are slightly less bent than those with ADP (Fig. 2, Fig. 6a, b and Supplementary Movie 1). The net bending between subunits M2-M5 is ~1.5° in the ADP-BeF$_x$ branch compared with ~2.1° in the ADP branch (Fig. 2c, d, h and Fig. 6c, d). The estimated bending values are 0.6°, 0.7°, and 0.2° for subunits M2-M5. Accordingly, the corresponding total strain energy in the bent segment of the mother filament in ADP-BeF$_x$ branch complex (0.6 $k_BT$) is slightly less than the ADP branch complex (0.9 $k_BT$).

**Mother filament twisting.** Like the ADP sample, the mother filament in the ADP-BeF$_x$ branches is over-twisted along with bending (Fig. 2j), at subunits M2-M4. Overtwisting is less pronounced (0.6°) in the ADP-BeF$_x$ branch compared with 0.9° in the ADP branch. We also observed this trend in bare, curved actin filaments[20] (Fig. 2k). Thus, ADP actin filaments are more compliant in both bending and twisting than ADP-BeF$_x$ filaments.

**Mother filament interactions.** The buried surface area between Arp2/3 complex and mother filament subunits is the same for branches with ADP and ADP-BeF$_x$ but the contacts of Arp3 are distributed differently on M4 and M3. In the branch junction with bound ADP contacts of Arp3 with the mother filaments bury 187.4 Å$^2$ on actin subunit M4 and 393.5 Å$^2$ on actin subunit M3, a total of 581 Å$^2$. In the branch junction with ADP-BeF$_x$ the total surface area buried between Arp3 and the mother filament is the same (589 Å$^2$), but the Arp3−M4 contact buries a total surface area of 202.3 Å$^2$ and Arp3−M3 contact buries 385.6 Å$^2$. This redistribution is due to the flatter conformation of Arp3 with ADP-BeF$_x$, which keeps the Arp3 outer domain (subdomains 1 and 2) closer to the mother actin subunit M4, while the inner domain (subdomains 3 and 4) is slightly further from the M3 subunit (Figs. 4f and Fig. 6).

**Daughter filament interactions.** Arp3 buries slightly more surface area (1524 Å$^2$) on daughter filament subunit D1 with ADP-BeF$_x$ bound to both Arp3 and actin D1 than with bound ADP (1415 Å$^2$). This larger contact area is largely attributable to the flatter conformation of the outer domain of Arp3 with bound ADP-BeF$_x$ that buries 1017 Å$^2$ on the outer domain of actin D1, compared with 912.4 Å$^2$ in the ADP-state. The difference in surface area is associated with specific sidechain and backbone interactions at the daughter filament interface (Fig. 3). With ADP-BeF$_x$ a hydrogen-bonding network stabilizes the D1−Arp3 interface (Fig. 3a, b) that is lost due to twisting of Arp3 without phosphate. R177 in the Arp3 W-loop hydrogen-bonds with the backbone of M47 in the D-loop of daughter filament subunit D1 (Fig. 3b), which is supported by a neighboring intra-subunit hydrogen-bond interaction between the backbone nitrogen of G48 and the sidechain oxygen of Q49 (Fig. 3b). These interactions promote contact between Q49 in the D1 subunit D-loop and the backbone oxygen of G199 in Arp3 (Fig. 3b). Table. 1

### Mechanical coupling between Arp2 and Arp3 mediated by ARPC3.

Subunits ARPC2, ARPC4 and ARPC5 are fixed on the mother filament in the ADP-BeF$_x$ and ADP structures, but Arp3 twisting after phosphate dissociation propagates changes to Arp2 and ARPC3 (Fig. 6a–d). Arp2, Arp3 and ARPC3 undergo a concerted hinge-like movement (Fig. 6c, d). ARPC3 moves with the Arp3 D-loop as it repositions away from the mother filament (Fig. 6c, d) inducing a 1° clamshell opening of Arp2 that shifts the Arp2 D-loop away from the mother filament (Fig. 6d, Supplementary Movie 1) and pivots the daughter filament by a

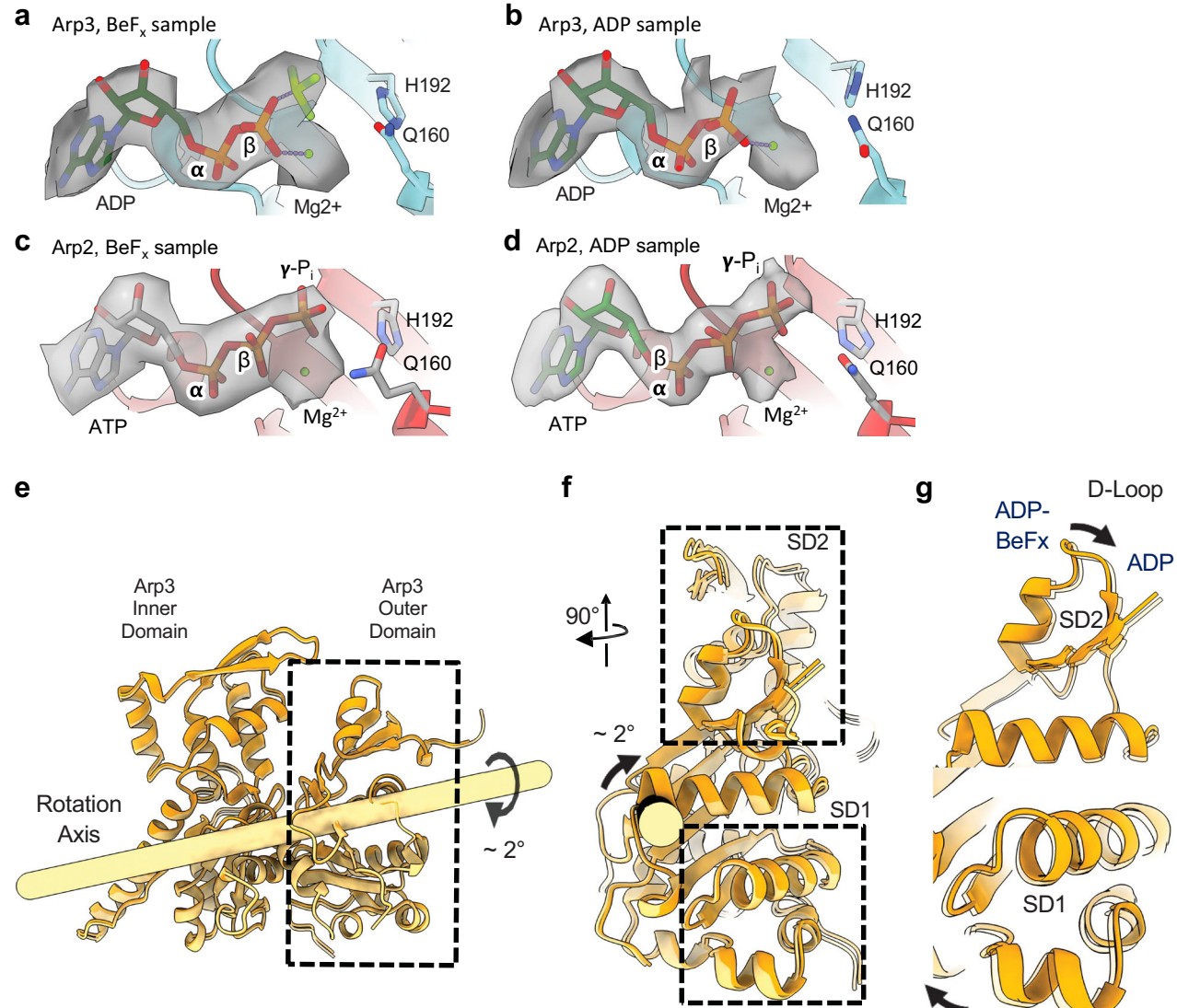

**Fig. 4 | The conformation of Arp3 in the branch junction depends on the bound nucleotide. a–d** Electrostatic potential densities, and nucleotide models in the nucleotide pockets of Arp3 and Arp2 with stick diagrams of the catalytic glutamine and histidine. **a** Arp3 with $BeF_x$ showing density for ADP, $BeF_x$ and $Mg^{2+}$. **b** Arp3 without $BeF_x$ has no density for a γ-phosphate. **c–d** Maps of Arp2 (from ADP-$BeF_x$ and ADP states, respectively) have density for ATP including the γ-phosphate and $Mg^{2+}$. **e** Ribbon diagram of Arp3 with a bar showing the axis of rotation between the inner and outer domains. The curved arrow indicates the motion of the outer domain when the inner domain is held fixed in space. **f–g** Comparison of the conformations of Arp3 in the ADP-$BeF_x$ (orange opaque) and ADP-bound (orange transparent) structures. Arp3 is flatter with bound $BeF_x$ because subdomains 1 (SD1)

and 2 (SD2) are rotated ~2° relative to the ADP-bound structure. The rotation axis in **e** and **f** was computed with UCSF ChimeraX as follows: two copies of the Arp3 model from our ADP branch structure were fitted by least-squares alignment to the Arp3 model in our ADP-$BeF_x$ branch structure. One ADP Arp3 model was fitted by subdomains 3 and 4, while the other was fitted by subdomains 1 and 2. The geometric transformation between the two ADP structures was then obtained by the 'measure rotation' command in UCSF ChimeraX. This procedure estimates the rotation axis together with the rotation angle and shift along the axis needed to superimpose subdomains 1 and 2 (which includes the D-loop) while holding subdomains 3 and 4 fixed.

small but measurable amount (0.2°) with respect to the mother filament (Fig. 6b, d; Supplementary Movie 1).

## Discussion

Previous work established that hydrolysis of ATP and dissociation of phosphate from actin filaments and Arp2/3 complex in branch junctions influences their stability under mechanical force[7]. Our two structures provide clues about the mechanisms. The high-resolution structure of the branch junction with ADP bound to the actin subunits and the Arps completes the inventory of interactions between Arp2/3 complex and the mother and daughter filaments and serves as the reference standard for other biochemical states. The second structure with $BeF_x$ substituting for phosphate to mimic the ADP-$P_i$

intermediate reveals subtle differences in the interactions of ADP-$BeF_x$-Arp3 with the mother and daughter filaments that may stabilize the branch.

The observed bending between subunits of the mother filament in contact with Arp2/3 complex in the branch junction (Fig. 2) indicates that local filament bending is linked to interactions with Arp2/3 complex. Mother filament bending is evident in two earlier structures of branch junctions[8,9] (Fig. 2), though it was not reported. Due to conservation of energy and detailed balance, if Arp2/3 complex bends filaments then it binds preferentially to bent filaments rather than straight ones. This explains why convex bends are favored for initiation of branches[21]. Either physical forces or thermal motions can create gentle bends in ADP- or ADP-$P_i$ filaments with surfaces complementary

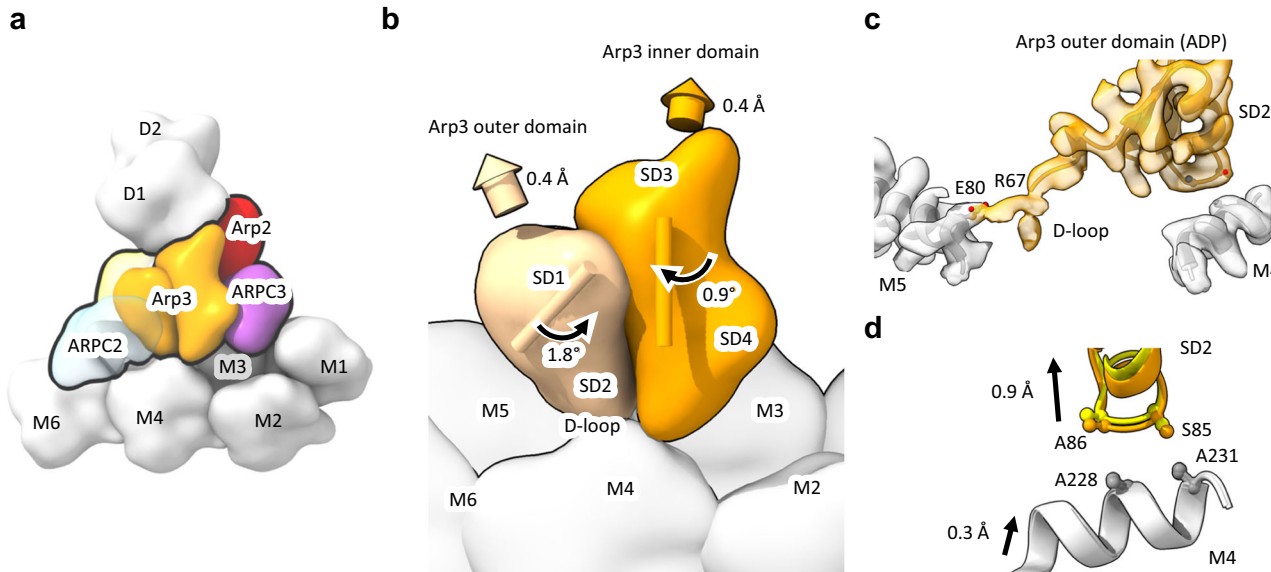

**Fig. 5 | Models comparing interactions of Arp3 with the mother filaments in branch junctions with bound ADP or ADP-BeFₓ. See also Supplementary Movie 1. a** Low-resolution rendering of the interaction of Arp3 with the mother filament in our atomic model of the ADP branch junction. **b** Rendering of Arp3 on the mother filament with subdomains marked SD1-SD4 as in Fig. 4. The arrows and numbers indicate the rotations and center of mass displacements of the Arp3 outer (pale orange isosurface) and inner (orange isosurface) domains with respect to the mother filament (light gray) going from the ADP-BeFₓ to the ADP-state. The D-loop in SD2 of Arp3 is farther from the mother filament with bound ADP than ADP-BeFₓ owing to a combined 1.8° rotation (left curved arrow) and 0.5 Å translation (upper left 3D arrowhead) of the Arp3 outer domain with respect to the mother filament. Colored cylinders show the axes of inner/outer subdomain rotation with respect to the mother filament (calculated in a similar manner to Fig. 4e–g). $\Delta\Delta\Theta$ and $\Delta\Delta T$ are the end-to-end bending and twisting angle differences between ADP and ADP BeFx from Fig. 2h, j. **c** Detail of the Arp3 SD2/D-loop region indicating contacts between SD2 and mother filament subunits M4 and M5. Unmodeled density corresponding to the Arp3 D-loop extends toward and makes contact with M5, while the globular portion of SD2 makes contact with M4. **d** A hydrophobic contact between SD2 and M5 loosens in the transition from ADP-BeFₓ to ADP branch complex structures. The alpha helix containing A228 and A231 in M5 follows the movement of SD2, but moves only 0.3 Å; Correspondingly, the distances between these residues and the Arp3 loop containing S85, A86 in increases by ~0.5 Å. Low-resolution isosurface representations (20 Å resolution) in **a** and **b** were generated by the UCSF ChimeraX 'molmap' command.

to Arp2/3 complex. Arp2/3 complex is positioned exactly the same with respect to the curved filament path, but the bend is slightly less with ADP-BeFₓ bound to the filament and Arp3 than with bound ADP. This difference in curvature could reflect either or both the effect of bound nucleotide on the conformation of Arp2/3 complex, or the higher stiffness of the mother filament with bound ADP Pᵢ[17]. We note, however, that the stabilizing effects of BeFₓ on filament debranching[7] originate from its binding Arp3 rather than the mother filament, since the mechanical stability of branches under force was identical with ADP- and ADP-BeFₓ- mother filaments[7].

The mother filament also twists locally where Arp2/3 complex binds (Fig. 2j, Supplementary Fig. 2). Concomitant twisting with bending is anticipated given the coupling between filament bending and twisting deformations (twist-bend coupling elasticity) predicted from theory and mathematical modeling of actin filaments[19]. The structures of mother filaments with bound Arp2/3 complex reported here support the existence of actin filament twist-bend coupling, as do the structures of bent filaments[20]. However, the reported extent and character of filament twisting for a given bending deformation in bare filaments[20] differs substantially from what we measured for branches (Supplementary Fig. 2c–f).

To assess if this difference originates from Arp2/3 complex binding or the analysis methods used to calculate the filament twist, we applied our analysis methods to bare, bent filaments[20]. Reanalysis with our method yielded a modest twist change of -0.1-0.4°(Fig. 2k, Supplementary Fig. 2b) similar to what we found in mother filaments bound to Arp2/3 complex (-0.4-0.6°; Fig. 2j, Supplementary Fig. 2A), rather than large twist oscillations (+/− 6°) between adjacent subunits[20] (Supplementary Fig. 2c). We are unsure of the precise origins of this difference (see Supplementary Information) but note that inter-

subunit twisting fluctuations of 6° would be associated with torsional strain energies of ~20 $k_B T$[15], nearly half the energy needed to instantly fragment an actin filament[22]. Nevertheless, the similar twist behavior obtained by our analysis for actin filaments, bent either spontaneously by thermal energy or by Arp2/3 complex binding (Fig. 2j, k), favors a mechanism in which mother filament twisting at branches originates from twist-bend coupling, an intrinsic property of the actin filament[19].

Our 2.7 Å resolution structure of the Arp2/3 branch junction with ADP bound to the actin subunits and Arp3 in Arp2/3 complex reveals several functionally important interactions of Arp2/3 complex with mother and daughter filaments that were not observed at lower resolution. The ARPC1 insert helix interacts with the mother filament through a hydrogen-bond and hydrophobic contacts. The ARPC2 C-terminal helix and tail bridge Arp2 and mother filament subunits through salt bridge and backbone hydrogen bonds. In addition, the ARPC4 C-terminal helix primarily contacts M5 subunit through salt bridges. Collectively, three major interfaces stabilize the binding of Arp2/3 complex to the mother filament through specific salt bridge, hydrogen bond, and hydrophobic interactions.

Branch formation involves activation of free Arp2/3 complex by binding nucleation promoting factors, which favors binding of Arp2/3 complex to the side of a mother filament and the conformational changes that allow Arp2 and Arp3 to nucleate a daughter filament[9,13]. As the Arps are brought together like two subunits along the short pitch helix of an actin filament, both change from a twisted to a flattened conformation, similar to subunits in actin filaments, and one or both hydrolyze their bound ATP. The Arps are flattened by a rotation of ~24° of their outer domains comprised of SD1 and SD2 relative to the inner domains. Flattening of Arp3 allows the D-loop in

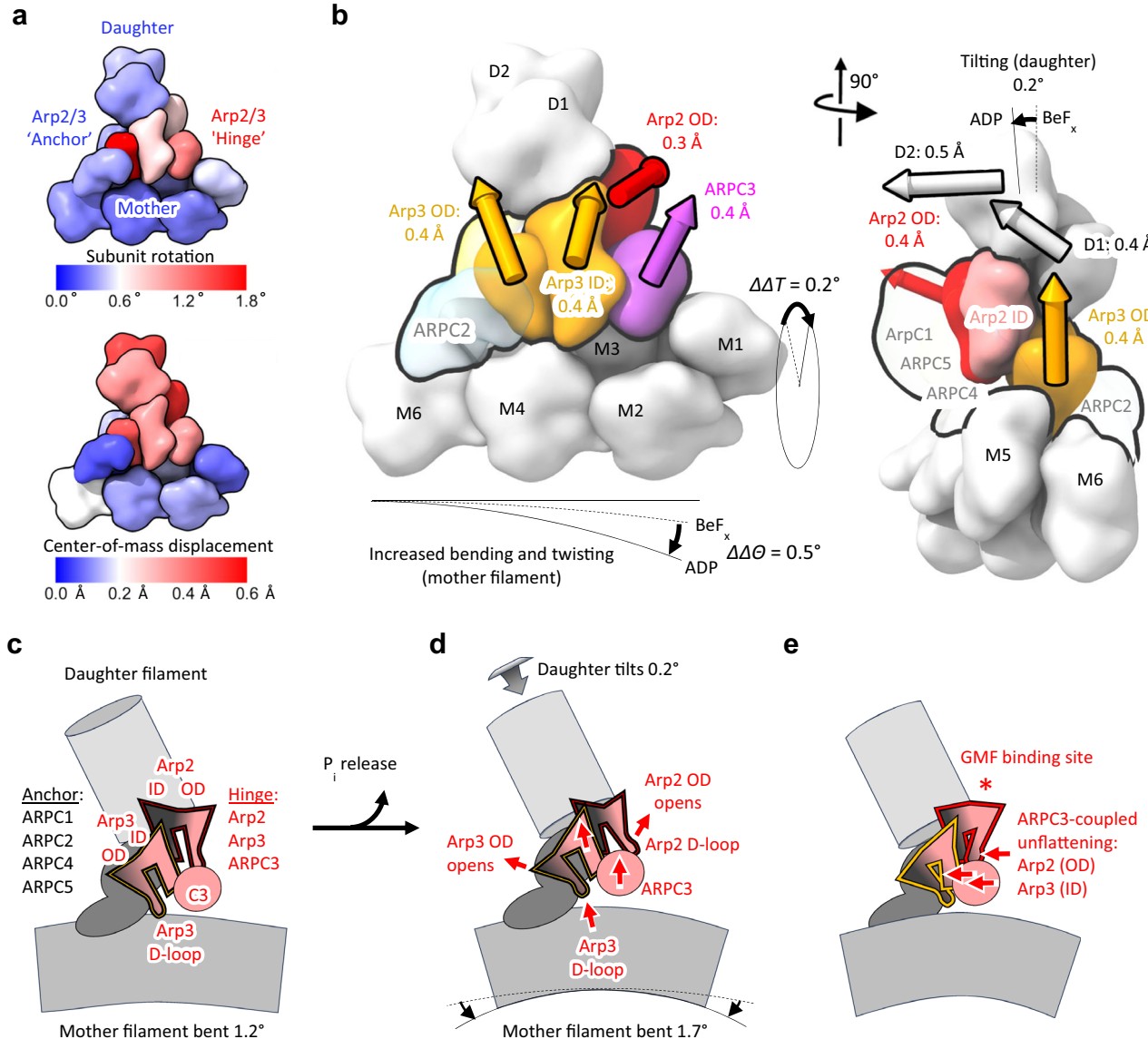

**Fig. 6 | Illustrations of subtle changes in branch conformation associated with phosphate release from Arp3. a**, **b** Diagrams depicting nucleotide-dependent, rigid-body rotation (**a**, top) and translation (**a**, bottom) magnitudes within branch junctions in the transition from ADP-BeF$_x$ to ADP structures (see also Supplementary Movie 1). Rotations are minimal in the mother filament, but increase significantly in Arp2/3 complex 'hinge' subunits, Arp2, Arp3, and ARPC3. Translations follow a similar pattern but are larger in the daughter filament due to tilting originating in the hinge. **b**, Subdomain translations with respect to the mother filament (colored 3D arrows; depicted arrow length is 100x the translation magnitude). ΔΔΘ and ΔΔT are the end-to-end bending and twisting angle differences between ADP and ADP BeF$_x$ from Fig. 2h, j. **c**–**e** Branch junction schematics illustrating differences in the structures with bound ADP-BeF$_x$ and ADP. **c** ADP-BeF$_x$

branch junction. **d** ADP branch junction. During phosphate release (C → D) an 'anchor' region (dark gray) consisting of subunits ARPC2, ARPC4 and ARPC5 remains mostly stationary on the mother filament as twisting of Arp3 moves it away from the mother filament. A concerted rigid body rotation moves ARPC3 against the Arp2 D-loop. The 'anchor' region constrains the inner domain of Arp2, so it undergoes a clamshell opening towards the conformation seen in the Dip1 complex[9]. **e** Schematic depicting large scale motions predicted by modeling completely un-flattened Arp2 and Arp3 in the branch junction (Supplementary Movie 2). The red asterisk denotes the GMF binding site on Arp2 identified by X-ray crystallographic studies[29]. OD, outer domain (subdomains 1 and 2); ID, inner domain (subdomains 3 and 4). Supplementary Fig. 4 gives a complete list of rotation and translation values calculated for the ADP-BeF$_x$ to ADP transition.

SD2 to interact with mother actin subunits M4 and M3 and is associated with a conformation change in the W-loop (residues 169-180) that not only opens a hydrophobic pocket for binding the daughter filament actin subunit D1 (Fig. 6) but also interferes with binding of C-helix of VCA[8], as also seen in Arp2/3 complex activated by Dip-1[9]. The dissociation of the VCA C-helix from the Arp3 allows the D1 actin subunit to bind Arp3.

Dissociation of the γ-phosphate of ATP leaves ADP in the active site of both Arps in the branch junction. The rates of hydrolysis and phosphate dissociation are not known but differ depending on the

source of Arp2/3 complex. Branches formed by mammalian Arp2/3 complex have ADP bound to both Arps[9,13], while branches formed by *S. pombe* Arp2/3 complex have ADP bound to Arp3 and ATP bound to Arp2[8]. The structures of branches with ATP or ADP-P$_i$ bound to both Arps are not available but are of great interest, since freshly formed branches are more mechanically stable than aged branches[7]. Therefore, we used BeF$_x$ to mimic the ADP-P$_i$ intermediate of *S. pombe* Arp2/3 complex.

Our structure of the branch junction with BeF$_x$ bound to Arp3 and ATP bound to Arp2 revealed that phosphate release from Arp3 is

**Table 1 | Cryo-EM data collection, refinement and model validation statistics**

| | ADP-State | ADP-BeF$_x$-State |
|---|---|---|
| **Data collection statistics** | | |
| Magnification | 64,000 | 64,000 |
| Voltage (kV) | 300 | 300 |
| Electron exposure (e⁻Å²) | 52.0 | 52.0 |
| Pixel size (Å) | 1.364 | 1.364 |
| Defocus range (μm) | −2.5–1.2 | −2.5–1.2 |
| Micrographs collected | 16,333 | 8519 |
| Tilt (°) | 0° | 0° and 20° |
| Number of particles | 406,334 | 183,479 |
| **Map Reconstruction Analysis** | | |
| Resolution (Å) at 0.143 threshold, CryoSPARC (masked/unmasked) | 2.73/2.98 | 3.22/3.90 |
| Map sharpening b-factor | −91.9 | −74.1 |
| **Model Refinement** | | |
| Initial model used (PDB) | 8E9B | 8E9B |
| Heavy atoms | 39,178 | 39,126 |
| Protein residues | 4938 | 4931 |
| Ligands (ATP/ADP/Mg²⁺/BeF$_x$) | 1/9/10/- | 1/9/10/7 |
| RMSD bond length (Å) | 0.004 | 0.004 |
| Bond Angles (°) | 0.68 | 0.73 |
| MolProbity score | 1.61 | 1.48 |
| Clash score | 5.9 | 6.6 |
| Ramachandran (Favored/allowed/outliers %) | 98.0/2.0/0.0 | 97.3/2.6/0.08 |
| Poor rotamers (%) | 2.7 | 2.5 |
| CaBLAM outliers (%) | 0.84 | 1.11 |

accompanied by a 2° rotation of the Arp3 outer domain, which reduces its flattening. While this structural change is subtle (Fig. 5d–f and Supplementary Movie 1), it supports models in which phosphate release compromises interactions of Arp3 with both mother and daughter filaments that contribute to the lower mechanical stability of mature branches with bound ADP[7]. We note that there is some ambiguity as to the nucleotide state of Arp2 in our structures. While there is density at the Arp2 gamma phosphate site in both ADP and ADP-BeF$_x$ branch structures, this density in the ADP-state structure seems to be weaker than would be expected for ATP, suggesting that ATP in the Arp2 site may be partially hydrolyzed[8] or, in the presence of BeF$_x$, partially substituted by this nucleotide analog. Further experiments are required to resolve this question.

The conformational changes observed here may be general and apply to the Arp2/3 complex from other organisms. The nucleotide-dependent structural changes reported here for *pombe* Arp2/3 complex result from hydrolysis and phosphate release in Arp3 only, but are concerted and long-range (e.g., ATPase of Arp3 the affects conformation of Arp2). Accordingly, hydrolysis and phosphate release by Arp2 could potentially induce additional long-range conformational changes in the complex, possibly explaining why the ATPase of Arp2 and/or Arp3 promotes dissociation of Arp2/3 complex branches in other organisms[5,13,23].

Implications for GMF-mediated debranching of actin filaments: The glial maturation factor (GMF) rapidly dissociates *S. pombe* ADP-Arp2/3 complex branches with half maximal concentration of 40 nM[7]. One the other hand, GMF concentrations up to 1 μM failed to dissociate branches formed by *S. pombe* Arp2/3 complex with bound ADP-BeF$_x$[7], suggesting weak binding of GMF to branches with bound ATP or ADP-P$_i$. Similarly, GMF has a higher affinity for mammalian ADP-Arp2/3 complex in solution than ATP-Arp2/3 complex[24].

Since BeF$_x$ binds and affects only Arp3 in the *S. pombe* complex, it is appealing to consider a mechanism in which GMF triggers debranching by preferentially binding the partially flattened ADP-Arp3 conformation. By analogy with cofilin binding to actin filaments[25–27], GMF binding is likely limited by site accessibility and conformational fluctuations of Arp3. These fluctuations presumably include 'unflattening' but must be larger than the 2° rotation observed here to account for the several order of magnitude change in GMF binding affinity[7], suggesting nucleotide-dependent, conformational dynamics of the Arp2/3 complex govern GMF binding. Also, by analogy with cofilin binding to actin filaments[28], subsequent GMF occupancy could favor conformational changes at the pointed end of Arp3 that weaken the Arp2/3 complex-mother interface and promote debranching.

This mechanism, while it ascribes significance to the structural changes of Arp3, is at variance with GMF binding Arp2 of unbranched *B. taurus* Arp2/3 complex in solution[29]. We note, however, that the structural changes originating in Arp3 observed here are equally consistent with a mechanism in which phosphate release from Arp3 promotes GMF binding to Arp 2 through linked conformational changes propagated from Arp3 to Arp2 through ARPC3. A concerted rigid-body rotation of Arp2 and ARPC3 triggered by Arp3 unflattening (Fig. 6c, d) slightly reorients the Arp2 D-loop, potentially facilitating GMF binding to the Arp2-D2 interface. Molecular modeling of completely un-flattened Arp2 and Arp3 conformations as seen in the GMF-Arp2/3 complex[29] suggest that ARPC3 mediated coupling between Arp2 and Arp3 are maintained for larger conformational changes, such as those associated with GMF binding to Arp2 (Fig. 6e and Supplementary Movie 2). Accordingly, GMF binding to Arp2 is expected to unflatten both Arp2 and Arp3, thereby weakening the mother and/or daughter filament interfaces and promoting debranching.

We cannot eliminate the possibility that GMF binds to different Arps in Arp2/3 complex depending on the organismal source and/or in solution vs in branches, nucleotide state(s), or that multiple GMF binding events (e.g., to Arp2 and Arp3) may promote debranching[30].

## Methods

Protein purification: Actin was purified from rabbit skeletal muscle acetone powder as described[7]. *S. pombe* Arp2/3 complex, GCN4-VCA and capping protein were purified as described[7,8]. Arp2/3 complex was stored in QB buffer (10 mM PIPES pH 6.8, 1000 mM NaCl, 1 mM MgCl₂, 1 mM EGTA, 0.1 mM ATP and 1 mM DTT)[8].

Assembly of branched junctions: Branched actin filaments were prepared as described[8]. Briefly, actin monomers with bound Ca²⁺ were converted to Mg²⁺-actin by equilibrating with 50 μM MgCl₂ and 0.2 mM EGTA (pH 7.5) for 10 min on ice. Actin was polymerized in the presence of capping protein (CP) by sequentially mixing 8.75 μM Mg-ATP-actin monomers with 0.75 μM CP and equilibrated at room temperature for 1 h. In parallel, Arp2/3 complex (0.4 μM) in QB buffer was activated by mixing 0.85 μM GCN4-VCA and 50 μM ATP and incubated at 4 °C for 1 h. The capped actin filaments sample was then gently mixed with an equal volume of activated Arp2/3 complex sample using cut pipette tips and equilibrated at 4 °C for 5 min. Daughter filaments were formed and elongated in the presence of CP by adding 0.25 μM Mg-actin monomers, 50 μM ATP, and 40 nM CP and incubated for 5 min at room temperature. This step was subsequently repeated 4 more times, then aged for ~90 min before preparing grids for cryo-EM data collection[7]. For Arp2/3 complex-ADP-BeF$_x$ samples, a final concentration of 2 mM BeF$_x$ (prepared using 2 mM BeSO₄ and 10 mM NaF)[7] was added during the 4ᵗʰ step of daughter filament formation (as mentioned above).

Sample Freezing and data collection: The samples were frozen on Quantifoil 1.2/1.3 300-mesh (Holey carbon) Au grids that were not glow-discharged. A sample of 3.0 μL was applied onto the carbon side of the grid using FEI Vitrobot™ Mark IV at 4 °C and 100% humidity. The samples were incubated on the grid for 50 s and the extra solution was blotted using two Vitrobot filter papers (Ø.55/20 mm, Grade 595, Ted

Pella) for 4 s at 0 blot force. The grids were plunged into liquid ethane at -180 °C with a wait time of 0.5 s. The vitrified grids were screened for sample homogeneity and ice thickness in a Glacios 200 kV transmission electron microscope equipped with Gatan K2 summit camera. Electron micrographs for image reconstructions were collected using Titan Krios equipped with X-cold field emission gun at 300 kV, Gatan image filter with slit width of 20 eV and a nanoprobe. A defocus value between −2.5 μm and −1.2 μm on a K3 Gatan summit camera in super-resolution mode was used to collect 1 movie per hole (using serialEM data collection software). Each movie contains 41 frames with a frame time of 0.08 s. A dose rate of 28.4 counts/pixel/s and a physical pixel size of 1.346 Å was used. In total, we collect ~16,000 movies of the ADP-state and 8519 movies of the ADP-BeF$_x$ state. Approximately half of the movies of the ADP-BeF$_x$ state were collected with the grids (alpha-tilt) normal to the beam and half with the grid tilted 20°.

EM Image processing: Both ADP and ADP-BeF$_x$ datasets were processed entirely using CryoSPARC v3[31]. Micrographs were subjected to motion correction (Patch Motion Correction) and CTF estimation before particle picking. About 3500 particles that contained a "Y-shaped" branch junction were picked manually using a box size of 256 Å. Manually picked particles included a variety of daughter filament orientations, such that their angles with respect to the optical axis ranged from 0 to 90 degrees. Special care was taken to include rare views where the daughter filaments were parallel (nearly parallel) to the optical axis. The manually picked particles were then subjected to 2-dimensional classification to create templates for template-based particle picking. The resulting template-based particle picking method (box size of 256 Å) identified ~3 million particles that contained junction particles, filament particles and other non-protein particles. These particles were subjected to another round of 2-D classification to separate junction particles from other filamentous particles. The resulting junction particle templates were subjected to iterative rounds of particle-picking and 2D-classification (Supplementary Fig. 5) to identify classes containing junction particles. The last iteration was followed by several more rounds of 2D-classification, yielding a total of ~1.7 million particles assigned as junction and filament particles. The final classification included several classes with daughter filaments that were aligned with the optical axis (2D classes, Supplementary Fig. 5) thus ensuring a complete 3D reconstruction without missing angle issues. These particles were then subjected to 3-dimensional structure refinement and classification using "Ab-initio 3D reconstruction" (10 classes) (Supplementary Fig. 5). For both ADP and ADP-BeF$_x$ samples, output structures from this 3D refinement/classification included a single junction class containing, ~400,000 junction particles (ADP-state) and ~200,000 particles (ADP-BeF$_x$ state). The particles were then subjected to homogeneous refinement, resulting in a 3.2 Å and 3.5 Å resolution map for the two samples respectively. This step was followed by CTF refinement (local and global), that further improved the resolution of the reconstruction to 3.0 Å and 3.3 Å respectively. Next, local motion correction was performed on the particles to further improve the resolution—which resulted in final reconstructions with resolutions of 2.7 Å for the ADP samples and 3.2 Å for the ADP-BeF$_x$ samples (Supplementary Fig. 6).

Model building and structure refinement: The map resolution in both states allowed us to build most residues and side chains unambiguously. The recent cryo-EM structure of the Arp2/3 complex in the branch junction at 3.5 Å resolution (PDB: 8E9B)[8] was used as the starting model for both structures. The atomic model was subjected to interactive all atom molecular dynamics flexible fitting run using ISOLDE[32]. Additional modeling was performed using COOT[33] and ISOLDE, followed a final refinement using Phenix[34]. Structures were validated using the "comprehensive validation (cryo-EM)" tool in Phenix[34]. Structural analysis (RMSD and buried surface area) and visualization was performed using ChimeraX[35]. Figures were generated using ChimeraX[35].

Reference frames for filament bending and twisting estimates: We utilize the methods and notation applied by Britton et al. to DNA[14], here adapted for arbitrary filaments such as actin. The reference frame for each filament subunit in a symmetric atomic filament structure (Fig. 2a, left side) is defined as follows: the origin $o$ lies on the helical symmetry axis, the unit vector **n** follows the helical symmetry axis and **s** points from $o$ toward the center of mass $c$ of the reference subunit. To track distortions in a helical structure, the atomic coordinates of the reference subunit are aligned by least-squares superposition to individual filament subunits $i$ (red arrows), carrying along the reference coordinate frame to define the subunit coordinate frame $o_i$, $s_i$, $l_i$, and $n_i$. In a distorted filament, deviations from symmetry cause misalignments between coordinate frames, as shown. This path definition allows the quantification of filament bending and twisting in the presence of other distortions like shearing between subunits[14]. Here, filament twist is defined as the axial rotation $T_i$ (red)[14] undergone between coordinate frames $i$ and $i+1$, taking into account any change in path direction that may occur (e.g., if $n_i$ is different than $n_{i+1}$); see Fig. 2 and Supplementary Fig. 2.

### Reporting summary

Further information on research design is available in the Nature Portfolio Reporting Summary linked to this article.

## Data availability

Atomic coordinates and corresponding cryo-EM density maps, including the half maps, masks and FSC curves used to estimate spatial resolution have been deposited in the Protein Data Bank (PDB) and Electron Microscopy Data Resource (EMD) under the accession codes 8UXW/ EMD- 42787 (ADP Arp2/3 branch complex), 8UXX/ EMD- 42788 (ADP BeF$_x$ Arp2/3 branch complex), 8UZ0/ EMD-42829 (straight F-actin from ADP Arp2/3 sample), and 8UZ1/EMD-42830 (straight F-actin from ADP BeF$_x$ Arp2/3 sample). Source data are provided with this paper.

## Code availability

A python script, generalized for arbitrary polymer atomic structures, was written for UCSF ChimeraX to perform filament twisting and bending analyses and visualizations in Fig. 5 and Supplementary Fig. 5. The script is publicly available on GitLab (https://gitlab.com/ cvsindelar/fil_twistbend3d). A second UCSF ChimeraX python script was written to generate domain rotation and shifting analyses, and associated visualizations in Figs. 5, 6 and S4. This script is also available on gitlab (https://gitlab.com/cvsindelar/subdomain_transforms).

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

## Acknowledgements

Research reported in this publication was supported by National Institute of General Medical Sciences of the National Institutes of Health under award numbers R35GM136656 to EMDLC, R01GM026338 to TDP, and R01GM110530 to CS. The content is solely the responsibility of the authors and does not necessarily represent the official views of the National Institutes of Health.

## Author contributions

S.S.C. designed experiments, prepared cryo-EM samples, collected data, performed 3D reconstructions and digital image data analysis, modeled and analyzed atomic structures and cowrote the first draft of the manuscript. S.Z.C. designed experiments, prepared samples, collected data and assisted with sample preparation. W.C. contributed to the twist/bend analysis and edited the manuscript. T.D.P. designed experiments, cowrote and edited the manuscript, and supervised the project. E.D.L.C. designed experiments, cowrote and edited the manuscript and supervised the project. C.V.S. designed experiments, modeled and analyzed atomic structures, modeled and analyzed structures, performed bending and twisting analysis, cowrote and edited the manuscript and supervised the project.

## Competing interests

The Authors declare no competing interests
