## [Peer Review File · Nature Communications]

Cryo-EM structures reveal how phosphate release from Arp3 weakens actin filament branches formed by Arp2/3 complexREVIEWER COMMENTS

Reviewer #1 (Remarks to the Author):

This work by Chavali et al., reports two high resolution structures of Arp2/3 complex mediated branched actin junction in ADP and ADP-BeFx bound states. The authors claim the ADP structure to be aged or mature branched junction and the ADP-BeFx bound structure to be in an ATP bound state. These structures are so far the highest resolution structures available till date and provide significant details about various interactions between Arp2/3 complex subunits, daughter and mother filament actin subunits. What is unique in these structures as compared to other Arp2/3 complex structures resolved at branched actin junctions is that the Arp2 subunit is in ATP bound state. This is quite surprising considering that all the other structures have shown hydrolysis of ATP in both the Arp subunits when Arp2/3 is incorporated into a branched junction. However, the authors fail to provide a clear explanation for this structural observation and attribute it to species differences. This study clearly demonstrates the conformational differences in Arp3 subunit between the ADP and the ADP-BeFx state (ATP bound state) that leads to slightly lesser interactions between Arp2/3 complex and the mother filament (based on the differences between the extent of buried surface areas). From these structures the authors show a subtle bending of the mother filament at the branched junction where the extent of bending is more for the mature (ADP-bound) state than the ADP-BeFx state. This is the first structural evidence of Arp2/3 interacting at bent actin junction as was proposed by Dan Fletcher's group. Although this is a higher resolution structure of Arp2/3 complex at branched actin junction, but there is no significant finding from this work as compared to what is already known in the field or from recent structural studies. Here are some of the major drawbacks of this study:

1. The authors claim that Arp2 is in ATP bound state in both aged /mature and newly formed branched junctions, but they fail to provide an explanation for the same. This is different from what has been observed in other structures of Arp2/3 complex at branched actin junctions. They speculate that this variation is due differences between vertebrate Arp2/3 complex and *S.pombe* Arp2/3 complex. They have not provided a direct experimental evidence for the same. In this manuscript there is no figure that compares the nucleotide states of Arp2 from the aged and the ADP-BeFx structures. Also, the only figure showing the nucleotide density in the Arp2 (Fig. 3C) is from the ADP-BeFx bound structure. It is unclear from the figure if the density is for ATP or for ADP-BeFx. Either one of them can be modelled or fitted into the nucleotide density at the reported resolution. The authors need to clarify the same and must provide experimental evidences to show species specific variation in nucleotide states of Arp2 at branched actin junction.

2. The authors show a difference in the bending of the mother actin filament at branch point between the aged (ADP-state) and the newly formed filament (the ADP-BeFx bound state). However, it is unclear if this is mediated by the difference in interactions of Arp2/3 complex with the mother filament in two different nucleotide states or due to the change in the nucleotide state of the mother filament itself

(ADP to ADP-BeFx state). It would have been better if the authors could have shown the same happening with other non-hydrolysable ATP analogue. While changing the nucleotide state of Arp2/3 complex, the author is also changing the nucleotide state of the actin filaments (both MF and DF). This makes this observation even more complicated and it fails to come to a definitive conclusion.

Overall it is an interesting manuscript that tries to explain the reason why an aged branched actin junction is less stable than a newly formed one, but there are no significant novel findings in this study from what was already known in the field. Although the high-resolution reconstructions provide more accurate atomic model of branched actin junction, but these structures do not provide any new or novel information than what was already known from other structural studies in the field.

Reviewer #2 (Remarks to the Author):

The branched actin network is important for many cellular activities including motility and membrane remodelling. The Arp2/3 complex is a conserved 7-subunit oligomer that is uniquely involved in nucleating new “daughter” actin filaments from existing “mother” filaments. In particular the whole Arp2/3 complex undergoes a conformational change on activation that brings its two actin-related Arp2 and Arp3 subunits into an F-actin-like short-pitch relationship. As with actin, Arp2 and Arp3 are ATP-binding proteins and ATP is a necessary cofactor for activation of the Arp2/3 complex. Also similar to actin dynamics, hydrolysis of bound ATP by Arp2 and Arp3 plays a role in branch turnover and overall function, but the mechanistic details of this process are not well understood.

Structure determination of Arp2/3 actin branches using cryo-EM has been particularly important in recent years, and Chavali use single particle cryo-EM to study Arp2/3 from *S. pombe*. The authors present two structures of reconstituted Arp2/3 branches with a focus on the nucleotide state of Arp3: 1) a higher resolution structure (nominal resolution, 2.7 Å) than the previously published Arp3-ADP branch (PMID 36442092) and 2) a lower resolution structure (nominal resolution 3.5 Å) of Arp3-ADP.BeFx branch. By comparison with the Arp3-ADP structure, the authors analyse the concerted conformational changes arising from Pi release within the complex.

Overall, these findings clarify details about subunit interfaces in the *S. pombe* Arp2/3 branch structure. In addition, the work provides new information about the subtle conformational changes associated with branch maturation. This insight does not require the higher resolution of Arp3-ADP reconstruction, but the small adjustments in the interaction between Arp2/3 and the mother filament could explain differences in branch stability seen on branch ageing. The methodology is mainly sound. This detailed analysis will be of interest to those focused on the molecular mechanisms of Arp2/3 and the manuscript

is written for such readers. A number of aspects relating to data presentation and analysis could be improved:

Major points

1. To provide additional support and validation for the reported higher resolution of the new Arp3-ADP reconstruction presented, and as is standard in the field, the corrected FSC curves with noise-substitution applied and model-map FSC curves and local resolution estimation should be included in Fig. S2.
2. The authors provide a detailed description of the new features of the Arp2/3 complex provided by their higher resolution structure (p3-4, Fig 2) but it is not clear what new mechanistic features arise from these details. This needs to be explained.
3. Relatedly, it is not clear what the overlaid presentation of new and previous structures in Fig 1B adds to the analysis. Without carefully reading the figure legend, it is not obvious that two structures are depicted.
4. The descriptions accompanying the structural analyses in Fig 3 and Fig 4 are not well explained. How was the rotation axis defined? How was the angle defined? How were the structures aligned?
5. As noted above, the differences between the Arp3-ADP.BeFx and Arp3-ADP structures are subtle, which makes their presentation challenging. The authors deal with this in Fig. 5A and Sup Movie 1 by extrapolating those changes 5-fold. Although the authors are relatively transparent about their approach in the legends of this figure/movie, this crucial information is not stated in the text and needs to be clearly explained to avoid being misleading about the scale of the changes observed. In particular in Fig. 5A, the authors should find a better way to show the concerted but subtle movements, for example by colouring the protein model by RMSD.
6. The authors should include a comparison of the nucleotide-dependent conformational changes in Arp3 with those described recently between ADP-Pi-F-actin and ADP-F-actin.
7. The authors should include some analysis of whether the conformational changes observed are expected to be general for Arp2/3 in other organisms and why. These latter 2 discussion points may be of more general interest than the detailed discussion of possible GMF debranching mechanisms currently included on p9-p10.

Minor points

1. The manuscript needs more careful proofreading. Multiple typos were spotted in Fig2 and FigS4. For example, the figure legend of 2B and 2C should be swapped; in the current Fig2B legend, "M5" should be "M3"; in FigS4B the actin should be labelled "M3"; in FigS4C the actin should be labelled "M3"; in FigS4D the actin should be labelled "M5"; in FigS4E the actin should be "M5".
2. Please show complete data collection, map reconstruction and model refinement statistics in Table 1. The map B factor, Rotamer outliers are needed.
3. Why doesn't the tight mask FSC curve for ADP-BeFx reconstruction in Fig S2 reach zero?

Response to review

In response to review, we carried out new work and revised many figures and the associated text. We believe that these revisions greatly strengthen the paper. We thank the reviewers for their helpful suggestions. For convenience and due to the number of significant changes to our manuscript, we attach a summary of revisions at the end of this document.

Reviewer #1 (Remarks to the Author):

This reviewer had both supportive and negative opinions about the manuscript. We combined their comments on each topic to facilitate our responses.

Positive comments

“This work by Chavali et al., reports two high resolution structures of Arp2/3 complex mediated branched actin junction in ADP and ADP-BeFx bound states.” “These structures are so far the highest resolution structures available till date and provide significant details about various interactions between Arp2/3 complex subunits, daughter and mother filament actin subunits.”

Response: Thank you for recognizing the value of the higher resolution structures.

This study clearly demonstrates the conformational differences in Arp3 subunit between the ADP and the ADP-BeFx state (ATP bound state) that leads to slightly lesser interactions between Arp2/3 complex and the mother filament (based on the differences between the extent of buried surface areas). From these structures the authors show a subtle bending of the mother filament at the branched junction where the extent of bending is more for the mature (ADP-bound) state than the ADP-BeFx state.

Response: Thank you for recognizing that the two new structures provide the first structural information on why aged branches are more sensitive to mechanical forces.

This is the first structural evidence of Arp2/3 interacting at bent actin junction as was proposed by Dan Fletcher's group.

Response: We revised the abstract and discussion to add this valuable point, thanks.

Misunderstandings

1. “The authors claim the ADP-BeFx bound structure to be in an ATP bound state.”

Response: The paper consistently explains that ADP-BeFx is an analog of the ADP-Pi state. We do not understand how the reviewer concluded that it is the ATP state.

2. “The authors claim the ADP structure to be aged or mature branched junction.”

Response: We do not understand why the reviewer questions whether the ADP-state corresponds to a mature branch junction.

Negative comments

1. The new structures do not advance our understanding of branch junction formation and dissociation. “There is no significant finding from this work as compared to what is already known in the field or from recent structural studies.” “Overall it is an interesting manuscript that tries to explain the reason why an aged branched actin junction is less stable than a newly formed one, but there are no significant novel findings in this study from what was already known in the field. Although the high-resolution reconstructions provide more accurate atomic model of branched actin junction, but these structures do not provide any new or novel information than what was already known from other structural studies in the field.”

Response: The only other high resolution structures of branch junctions are those with mammalian Arp2/3 complex at 3.9 Å resolution by Ding et al. (2022) and S. pombe Arp2/3 complex at 3.5 Å resolution by Chou et al. (2022). Our manuscript redefines the positions and orientations of amino acid side chains not well resolved at lower resolution in either of these papers. Many of these residues are located at interfaces between Arp2/3 complex and the mother filament and are therefore relevant to the stability of the branch junction.

More importantly, neither of those papers investigated the conformational change and destabilization associated with phosphate release, which the reviewer acknowledges as valuable.

2. Concern about the nucleotide bound in Arp2. “What is unique in these structures as compared to other Arp2/3 complex structures resolved at branched actin junctions is that the Arp2 subunit is in ATP bound state.” “This is quite surprising considering that all the other structures have shown hydrolysis of ATP in both the Arp subunits when Arp2/3 is incorporated into a branched junction.” “The authors claim that Arp2 is in ATP bound state in both aged /mature and newly formed branched junctions, but they fail to provide an explanation for the same. This is different from what has been observed in other structures of Arp2/3 complex at branched actin junctions. They speculate that this variation is due differences between vertebrate Arp2/3 complex and S. pombe Arp2/3 complex.”

Response: Of the two published, high-resolution structures of branch junctions, the one with mammalian Arp2/3 complex (Ding et al. 2022) has ADP bound to both Arps after 4 hours of ageing. The branch in the Chou et al. (2022) paper with S. pombe Arp2/3 complex was aged for 80 min. The authors’ interpretation of the density in the active site of Arp2 was that the gamma-phosphate site was ~70% occupied, so hydrolysis and phosphate dissociation occurred, but slowly. The higher resolution structure in the current paper confirms that interpretation. Additional structures will be required to determine if the difference is due to sample preparation or the source of Arp2/3 complex.

3. “The authors fail to provide a clear explanation for this structural observation and attribute it to species differences.” “The authors need to clarify the same and must provide experimental evidences to show species specific variation in nucleotide states of Arp2 at branched actin junction.”

Response: Chou et al. (2022) copied below explain what is known about ATP hydrolysis during branch formation. However, note that next to nothing is known about phosphate dissociation, the issue regarding the presence of phosphate in the active site of Arp2 in the S. pombe branch junctions. Analyzing nucleotide hydrolysis and phosphate dissociation in two species are far beyond the scope of this structural study.

From Chou et al. (2022):

“The branch structure shows directly that Arp2 and Arp3 hydrolyze their bound ATP at substantially different rates. Like all seven actin subunits in the structure, Arp3 hydrolyzed its bound ATP and dissociated the γ -phosphate during the 80 min required for specimen preparation to leave just ADP in the active site. On the other hand, Arp2 retained most of the γ -phosphate, so hydrolysis and/or phosphate release was slower in spite of the catalytic residues in the active site being positioned to promote hydrolysis. ATP is bound to both Arps in the complex with Dip1 in spite of nucleating a daughter filament (6).

Our observation of the γ -phosphate in the active site of Arp2 in the branch junction is surprising given previous biochemical analysis and mutations of the active site. Dayel et al. (35) found that hydrolysable ATP bound to Arp2/3 complex is required for branch nucleation, although the ATP analog may have influenced the observations. Both Dayel et al. (36) and Martin et al. (37) reported that Arp2 hydrolyzes its bound ATP coincident with actin filament nucleation. Martin et al. (38) found that mutations of residues in the active site of budding yeast Arp2 and Arp3 can compromise ATP binding and cause defects in clathrin-mediated endocytosis. However, mutations compromising ATP hydrolysis did not prevent budding yeast (38) or *Drosophila* (39) Arp2/3 complex from nucleating actin assembly but stabilized lamellipodial actin filament networks.

These differences from the nucleotides in the branch junction suggest that new work is needed on the order and rates of these events: formation of a short-pitch Arp2/Arp3 dimer; flattening of Arp2 and Arp3; delivery of actin subunits D1 and D2 from VCA to the Arps; dissociation of VCA; nucleotide hydrolysis; and phosphate release.”

4. “They have not provided a direct experimental evidence that Arp2 is in ATP bound state in both aged /mature and newly formed branched junctions). In this manuscript there is no figure that compares the nucleotide states of Arp2 from the aged and the ADP-BeF_x structures.”

Response: We added a panel to Fig 3 with the map and model of the Arp2 nucleotide binding pocket in the mature branch junction structure in the presence of BeF_x. The new panel shows clear and unambiguous density for gamma phosphate bound to Arp2 in both structures.

5. “The only figure showing the nucleotide density in the Arp2 (Fig. 3C) is from the ADP-BeF_x bound structure. It is unclear from the figure if the density is for ATP or for ADP-BeF_x. Either one of them can be modelled or fitted into the nucleotide density at the reported resolution.”

Response: We agree it is difficult to distinguish a gamma phosphate from BeF_x in cryo-EM maps. However, two lines of evidence support the interpretation that phosphate occupies the gamma-phosphate position of Arp2 in the ADP-BeF_x-structure.

First, at early stages of refinement of the BeF_x structure (lower resolution than the final structure), the map had density for gamma-phosphate in Arp2 but not for BeF_x bound to actin or Arp3. The BeF_x density only appeared at higher resolution. This effect was also seen in actin filament structures with BeF_x from the Raunser lab: the high resolution structure of Oosterheert et al., 2022, has density for this ligand that was missing in the lower-resolution structure of Merino et al., 2018.

Second, from a biochemical standpoint, it is difficult to reconcile why BeF_x would occupy the Arp2 active site in the BeF_x sample. Given that the gamma-phosphate site of Arp2 in the ADP sample is about 70% occupied by either the ATP gamma-phosphate or a free gamma-phosphate, it would be logical to

conclude the same in the BeF_x sample. The experimental conditions for the ADP and ADP-BeF_x samples are identical except for inclusion of BeF_x.

6. “The authors show a difference in the bending of the mother actin filament at branch point between the aged (ADP-state) and the newly formed filament (the ADP-BeF_x bound state). However, it is unclear if this is mediated by the difference in interactions of Arp2/3 complex with the mother filament in two different nucleotide states or due to the change in the nucleotide state of the mother filament itself (ADP to ADP-BeF_x state).” “While changing the nucleotide state of Arp2/3 complex, the author is also changing the nucleotide state of the actin filaments (both MF and DF). This makes this observation even more complicated and it fails to come to a definitive conclusion.”

Response: We thank the reviewer for this helpful comment. Additional analysis of our structures revealed more precisely the bending and twisting behavior of the mother filament bound to Arp2/3 complex +/- BeF_x. Bending is localized to actin subunits directly next to Arp3 and the bend is slightly less pronounced in the presence of BeF_x. While we attributed this bending difference to conformational changes in ADP BeF_x Arp2/3 complex, the nucleotide state of the mother filament could also contribute. For example, we note that if the mother filament is stiffer with ADP BeF_x than with ADP, then Arp2/3 complex binding could bend the mother filament bend less acutely. This possibility is highlighted by recent evidence that ADP-P_i actin is stiffer than ADP actin (Matthews et al., 2022 cited in the revised text). So it is unclear whether the difference of mother filament bending in these two nucleotide states comes from the Arps. We noted these possibilities in the revised manuscript.

In analyzing the twisting behavior for this revision, we also discovered evidence of twist-bend coupling as was reported in recent structures of bare actin filaments (Reynolds et al., Nature 611 :380). These new results are included with an additional supplementary figure and a brief discussion.

7. It would have been better if the authors could have shown the same happening with other non-hydrolysable ATP analogue.

Response: Branches do not form in the presence of nonhydrolyzable ATP analogs (e.g. AMPPNP; Pandit et al., PNAS 2020 www.pnas.org/cgi/doi/10.1073/pnas.1911183117). The paper considers relevant nucleotide states that have been biochemically characterized (ADP vs. ADP ±P_i (BeF_x)).

Reviewer #2 (Remarks to the Author):

The branched actin network is important for many cellular activities including motility and membrane remodelling. The Arp2/3 complex is a conserved 7-subunit oligomer that is uniquely involved in nucleating new “daughter” actin filaments from existing “mother” filaments. In particular the whole Arp2/3 complex undergoes a conformational change on activation that brings its two actin-related Arp2 and Arp3 subunits into an F-actin-like short-pitch relationship. As with actin, Arp2 and Arp3 are ATP-binding proteins and ATP is a necessary cofactor for activation of the Arp2/3 complex. Also similar to actin dynamics, hydrolysis of bound ATP by Arp2 and Arp3 plays a role in branch turnover and overall function, but the mechanistic details of this process are not well understood.

Response: Thank you for summarizing the contribution of this paper.

Structure determination of Arp2/3 actin branches using cryo-EM has been particularly important in recent years, and Chavali use single particle cryo-EM to study Arp2/3 from *S. pombe*. The authors present two structures of reconstituted Arp2/3 branches with a focus on the nucleotide state of Arp3: 1) a higher resolution structure (nominal resolution, 2.7 Å) than the previously published Arp3-ADP branch (PMID 36442092) and 2) a lower resolution structure (nominal resolution 3.5 Å) of Arp3-ADP.BeFx branch. By comparison with the Arp3-ADP structure, the authors analyse the concerted conformational changes arising from Pi release within the complex.

Overall, these findings clarify details about subunit interfaces in the *S. pombe* Arp2/3 branch structure. In addition, the work provides new information about the subtle conformational changes associated with branch maturation. This insight does not require the higher resolution of Arp3-ADP reconstruction, but the small adjustments in the interaction between Arp2/3 and the mother filament could explain differences in branch stability seen on branch ageing. The methodology is mainly sound. This detailed analysis will be of interest to those focused on the molecular mechanisms of Arp2/3 and the manuscript is written for such readers.

Response: Thank you again for this clear summary of the main findings.

A number of aspects relating to data presentation and analysis could be improved:

Major points

1. To provide additional support and validation for the reported higher resolution of the new Arp3-ADP reconstruction presented, and as is standard in the field, the corrected FSC curves with noise-substitution applied and model-map FSC curves and local resolution estimation should be included in Fig. S2.

Response: We added the requested information.

2. The authors provide a detailed description of the new features of the Arp2/3 complex provided by their higher resolution structure (p3-4, Fig 2) but it is not clear what new mechanistic features arise from these details. This needs to be explained.

Response: The high resolution ADP-structure fills in most parts of the branch junction missing at lower resolution. Many of these features participate in interactions between Arp2/3 complex and the mother filament, completing the account of the interactions that stabilize the branch junction. Comparing the ADP-BeFx-structure with the reference ADP-structure revealed subtle conformational changes that weaken the branch junction after phosphate release from Arp3. Many details including the change in interactions between Arp3 and daughter subunit D1 could not be appreciated at lower resolution. Detection of the bend in the mother filament adjacent to Arp2/3 complex explains why bent actin filaments are favored for binding Arp2/3 complex and initiating branches.

3. Relatedly, it is not clear what the overlaid presentation of new and previous structures in Fig 1B adds to the analysis. Without carefully reading the figure legend, it is not obvious that two structures are depicted.

Response: We thank the reviewer for pointing this out and agree with this assessment. We moved up Fig.2 to take the place of Fig. 1B, since this serves the intended function of showing key differences between the structures.

4. The descriptions accompanying the structural analyses in Fig 3 and Fig 4 are not well explained. How was the rotation axis defined? How was the angle defined? How were the structures aligned?

Response: We thank the reviewer for pointing this out. We remade Figures 3 and 4 (now Figures 4 and 5) to show the conformational changes more clearly. The expanded caption explains how the alignments were performed and how the rotation was defined.

5. As noted above, the differences between the Arp3-ADP.BeF_x and Arp3-ADP structures are subtle, which makes their presentation challenging. The authors deal with this in Fig. 5A and Sup Movie 1 by extrapolating those changes 5-fold. Although the authors are relatively transparent about their approach in the legends of this figure/movie, this crucial information is not stated in the text and needs to be clearly explained to avoid being misleading about the scale of the changes observed. In particular in Fig. 5A, the authors should find a better way to show the concerted but subtle movements, for example by colouring the protein model by RMSD.

Response: We thank the reviewers for this very helpful comment. We did a more careful investigation of subunit rotations and translations, and devised a way to quantitatively represent these in the new Fig. 5 (now renumbered to Fig. 6) .

6. The authors should include a comparison of the nucleotide-dependent conformational changes in Arp3 with those described recently between ADP-Pi-F-actin and ADP-F-actin.

Response: We thank the reviewer for this suggestion. While only minor conformational differences are observed between most actin filament nucleotide states, Oosterheert et al. 2022 reported that the actin C-terminus assumes a unique conformation in the ADP BeF_x nucleotide state. However, our review of their deposited data, the raw cryo-EM density did not support this conclusion, since the C-terminus was mostly disordered in this region of the maps. We also investigated inner/outer domain motions in the available structures of actin filaments in different nucleotide states. However, the relative rotations/translations of inner and outer domains were generally small and showed no clear trend. Thus, the interface of the pointed ends of Arp2 and Arp3 with the mother filament seems to allow more freedom of movement. Thus, no clear conclusions can be drawn comparing the response of Arp2 and Arp3 with actin to nucleotide state.

7. The authors should include some analysis of whether the conformational changes observed are expected to be general for Arp2/3 in other organisms and why. These latter 2 discussion points may be of more general interest than the detailed discussion of possible GMF debranching mechanisms currently included on p9-p10.

Response: We thank the reviewer for the suggestion. We added to the text a paragraph explaining how these conformational changes may be general and apply to Arp2/3 complex from other organisms:

“The conformational changes observed here may be general and apply to the Arp2/3 complex from other organisms. The nucleotide-dependent structural changes reported here for *pombe* Arp2/3 complex result from hydrolysis and phosphate release in Arp3 only, but are concerted and long-range (e.g., ATPase of Arp3 affects conformation of Arp2). Accordingly, hydrolysis and phosphate release by Arp2 could potentially induce additional long-range

conformational changes in the complex, possibly explaining why the ATPase of Arp2 and/or Arp3 promotes dissociation of Arp2/3 complex branches in other organisms^{5,13,22}.”

Minor points

1. The manuscript needs more careful proofreading. Multiple typos were spotted in Fig2 and FigS4. For example, the figure legend of 2B and 2C should be swapped; in the current Fig2B legend, “M5” should be “M3”; in FigS4B the actin should be labelled “M3”; in FigS4C the actin should be labelled “M3”; in FigS4D the actin should be labelled “M5”; in FigS4E the actin should be “M5”.

Response: Thank you for noting these typos, which we corrected.

2. Please show complete data collection, map reconstruction and model refinement statistics in Table 1. The map B factor, Rotamer outliers are needed.

Response: The requested information has been added

3. Why doesn't the tight mask FSC curve for ADP-BeFx reconstruction in Fig S2 reach zero?

Response: We thank the reviewer for pointing this out. After performing the more comprehensive FSC calculations suggested by the reviewer, it is clear that the non-zero signal is due to high resolution signal present at the Nyquist frequency in the ADP-BeFx reconstruction.

Summary of revisions

Abstract

- Added the results of the new analysis of mother filament bend and twist in branch junction

Results

- Added a new analysis of mother filament bending and twisting in the high resolution ADP-branch structure, including thermodynamics; new text and new Fig 2.
- Added a new comparison of bending and twisting of ADP-P_i- and ADP-actin filaments from ref 19. Confirmed overtwist but found that it is more than an order of magnitude less than reported.
- Added a new analysis of mother filament bending and twisting in the high resolution ADP-BeFx-branch structure, including thermodynamics; added new text and new Fig 2.

Discussion

- Found bends in the mother filaments in the previous, lower resolution structures of branch junctions.
- Used detailed balance to explain why bent filaments are favored for branch formation.

Figures

- Combined former figures 1 and 2 into new Fig 1.

- Added new Fig 2 (replacing original Fig S5) with the analysis of bending and twisting of mother filaments and bare actin filaments.
- Added a new panel on the active site in the ADP-BeF_x structure to Fig 3, now Fig 4.
- Made new renderings of the conformational changes associated with phosphate release in Arp3 in Fig 4, now Fig 5.
- Made new renderings of the conformational changes associated with phosphate release in the Arp2/3 complex, mother filament and daughter filament in Fig 5, now Fig 6.

Supplemental materials

- New Fig S2 with analysis of twisting in bare actin filaments
- New Fig S4 summarizing subunit rotations and translations between branch junctions structures with ADP-BeF_x and ADP.
- New Fig S6 (replacing original Fig S2) with new estimates of resolution of the two structures showing both FCS and map/model FSC calculations.
- New supplementary text explaining the filament twist analysis

REVIEWERS' COMMENTS

Reviewer #1 (Remarks to the Author):

The authors have made substantial changes to the paper after receiving the comments from the reviewers. Each critique has been thoroughly considered, and the authors have provided robust explanations for every point raised. While the paper could benefit from additional experimental evidence to bolster its claims, it significantly contributes to the actin field through its meticulous structural analysis. The presented evidence for the authors' claims is not only satisfactory but has been further fortified by the detailed structural analysis. There is no further objections for acceptance of the revised version of the manuscript for publication.

Reviewer #2 (Remarks to the Author):

The authors have addressed my technical issues. The manuscript necessarily remains focused on small structural changes within the Arp2/3 complex, the mechanistic significance of which are unclear, and the analysis is unlikely to be readily comprehensible to the broad readership of Nature Communications.